# Comparison of Clinical Outcomes between Robot-Assisted Partial Nephrectomy and Cryoablation in Elderly Patients with Renal Cancer

**DOI:** 10.3390/cancers14235843

**Published:** 2022-11-26

**Authors:** Shohei Kawaguchi, Kouji Izumi, Renato Naito, Suguru Kadomoto, Hiroaki Iwamoto, Hiroshi Yaegashi, Takahiro Nohara, Kazuyoshi Shigehara, Kotaro Yoshida, Yoshifumi Kadono, Atsushi Mizokami

**Affiliations:** 1Department of Integrative Cancer Therapy and Urology, Kanazawa University Graduate School of Medical Science, Kanazawa 920-8641, Japan; 2Department of Radiology, Kanazawa University Graduate School of Medical Science, Kanazawa 920-8641, Japan

**Keywords:** renal cell carcinoma, robot-assisted partial nephrectomy, cryoablation, oncologic outcome, renal function

## Abstract

**Simple Summary:**

Advances in diagnostic imaging have led to an increase in the diagnosis and treatment of small-diameter renal cell carcinomas in the elderly. Elderly patients may show impaired operative tolerance; thus, treatment should be more carefully chosen in them than in younger patients. In this study, a retrospective comparison of robot-assisted partial nephrectomy (RAPN) and percutaneous cryoablation (PCA) was conducted for small-diameter renal cell carcinomas in elderly patients. Both RAPN (with a slightly high complication rate but a low recurrence rate) and PCA (with a low complication rate but a slightly high recurrence rate) could be performed safely in elderly patients. RAPN proved to be a safe and effective method for treating small-diameter renal cell carcinomas in elderly patients, thereby being the first treatment of choice in such patients. PCA was also a safe and feasible alternative, especially in patients in whom general anesthesia poses a high risk.

**Abstract:**

Advances in imaging technologies have increased the opportunities for treating small-diameter renal cell carcinomas (RCCs) in the elderly. This retrospective study based on real-world clinical practice compared perioperative complications, preoperative and postoperative renal function, recurrence-free survival, and overall survival in elderly patients with RCC who had undergone robot-assisted partial nephrectomy (RAPN) or percutaneous cryoablation (PCA). A total of 99 patients (aged ≥70 years), including 50 and 49 patients in the RAPN and PCA groups, respectively, were analyzed. In the entire cohort, Clavien–Dindo grade ≥3 complications occurred in only one patient who had undergone RAPN. Renal function was significantly lower in the postoperative period than in the preoperative period in both the RAPN and PCA groups. The recurrence-free survival and overall survival rates were worse in the PCA group than in the RAPN group, albeit not significantly. RAPN was considered a safe and effective method for treating RCCs in elderly patients. Moreover, although the recurrence rate was slightly higher in the PCA group than in the RAPN group, PCA was deemed to be a safe alternative, especially for treating patients in whom general anesthesia poses a high risk.

## 1. Introduction

Renal cell carcinoma (RCC) accounts for approximately 3% of all cancers, with an estimated peak incidence among those aged 60–70 years [1]. In recent years, the widespread use of high-resolution imaging techniques, such as computed tomography (CT) and magnetic resonance imaging (MRI), has led to an increase in the diagnosis of small, asymptomatic renal tumors [2,3]. Surgery remains the only curative treatment option for localized RCCs, and localized T1 RCCs are best managed with partial nephrectomy (PN) rather than radical nephrectomy (RN) [1]. A meta-analysis suggested that robot-assisted partial nephrectomy (RAPN), which is associated with lower estimated blood loss, shorter postoperative hospital stay, and fewer postoperative complications, might be more suitable for treating small-diameter RCCs than open nephrectomy [4].

Recently, local therapy has been recommended as an alternative approach in patients at risk for surgery, such as elderly patients and those with comorbidities [5]. While some studies suggest that RAPN and ablation therapy (AT) have similar oncological outcomes in patients with small-diameter RCCs [6], others report that the recurrence rate is significantly higher after AT than after RAPN [2,7], highlighting the need for appropriate treatment selection. Due to the increasing opportunities for treating small-diameter RCCs in the elderly, we conducted a retrospective study to compare the relationship of specific treatment approaches with clinical outcomes in real-world clinical practice.

## 2. Materials and Methods

This retrospective study included patients aged ≥70 years who were diagnosed with RCC and had undergone RAPN or percutaneous cryoablation (PCA) at Kanazawa University Hospital between October 2016 and December 2021. Preoperative contrast-enhanced CT and MRI were performed for staging, and tumor size and localization were evaluated using the R.E.N.A.L. score [8]. Tumors with a nephrometry score of 4, 5, or 6 points were considered to have low complexity; those with a score of 7, 8, or 9 points were deemed to have moderate complexity; and those with a score of 10, 11, or 12 points were considered to have the highest complexity [8]. In cases of renal dysfunction where the usage of a contrast medium was difficult, plain CT and MRI were used for evaluation. The indications for RAPN and PCA were determined based on the patient’s general condition, complications, general anesthesia tolerance, and patient preference.

RAPN was performed using the transperitoneal or retroperitoneal approach based on the location and size of the tumor and the anatomical location of the renal vessels. Tumor resection was performed with warm ischemia using a renal artery clamp. After tumor excision, hemostasis and urinary tract repair were performed with inner sutures and parenchymal sutures were placed as needed. PCA was performed under CT guidance using the CryoHit cryoablation system (Galil Medical, Yokneam, Israel) and 17-guage cryoneedles (IceRod; Galil Medical) with local anesthesia [9]. In patients with large or hypervascular tumors at a high risk of bleeding, arterial embolization with absolute ethanol and ethiodized oil (lipiodol; Guerbet Japan, Tokyo, Japan) was performed a day before PCA. Where possible, needle biopsy was simultaneously performed with PCA to confirm the histologic type of the tumor.

Postoperative local recurrence and distant metastasis were evaluated using thoracoabdominal contrast CT. In patients who were not suitable for imaging using contrast media due to severe renal dysfunction, recurrence was evaluated using plain CT or MRI. Renal function was assessed by determining estimated glomerular filtration rates (eGFR) at 1, 3, 6, 12, and 24 months after RAPN and PCA, and postoperative eGFR preservation was calculated using the following formula: % eGFR = postoperative eGFR / preoperative eGFR × 100 [10]. The quality of RAPN was assessed using the achievement of trifecta, which was defined as a negative surgical margin, absence of postoperative complications, and warm ischemia time of ≤25 min [11].

Chi-square, Wilcoxon’s signed-rank, and Mann–Whitney U tests were used to compare the features of RAPN and PCA groups. Odds ratios (ORs) with 95% confidence intervals (CIs) were used to determine the risk for a decline of ≥20% in % eGFR at postoperative year one. Multivariable logistic regression was performed to identify independent of variables. Variables with a *p* value of <0.1 in the univariable analysis were included in the multivariable analysis. Survival curves were measured using the Kaplan–Meier method, and differences in overall survival (OS) and recurrence-free survival (RFS) were evaluated using the log-rank test. All data analyses were performed using SPSS ver. 25.0 (SPSS, Chicago, IL, USA), and a *p* value of <0.05 was considered statistically significant.

## 3. Results

A total of 50 patients who had undergone RAPN and 49 patients who had undergone PCA at Kanazawa University Hospital were retrospectively analyzed. All analyzed patients were ≥70 years old. Table 1 shows the demographic characteristics and preoperative renal function of these 99 patients.. The median age was significantly higher in the PCA group than in the RAPN group (78 years vs. 75 years, respectively; *p* = 0.010). Although no significant differences in the incidence of hypertension, diabetes, and cardiovascular disease were observed between the two groups, cerebrovascular disease was significantly more prevalent in the PCA group than in the RAPN group (*p* = 0.010). The number of patients with a solitary kidney was significantly higher in the PCA group than in the RAPN group (8 vs. 1, *p* = 0.041). However, no significant difference in the preoperative eGFR was observed between the two groups.

Table 2 presents the reasons for choosing PCA in the study cohort. Ten patients were unable to tolerate general anesthesia and harbored absolute contraindications for RAPN. Seventeen patients harbored relative contraindications for RAPN due to a solitary kidney, previous surgery, renal vascular abnormalities, or difficulty in intraoperative tumor identification. Most patients had comorbidities, and PCA was performed in only three patients without comorbidities based on patient preference.

The tumor features are shown in Table 3. In brief, the maximum tumor diameter was 2.7 ± 1.2 cm in the RAPN group and 2.4 ± 0.8 cm in the PCA group, with no significant difference between the two groups. Most of the patients in both the groups had clinical Ta RCC, and the R.E.N.A.L. scores were low or moderate in most patients.

The surgical margins were positive in 2 (4.0%) patients, and trifecta was achieved in 36 (72.0%) patients. In the entire study cohort, Clavien–Dindo grade ≥3 complications were observed in only one patient (grade 3a pneumothorax) who had undergone RAPN. The mean follow-up period was 24.3 (2–60) months and 20.1 (2–60) months for the RAPN and PCA groups, respectively.

In the RAPN group, the preoperative mean eGFR was 65.0 ± 17.6 mL/min/1.73 m^2^ and the postoperative eGFR was significantly lower than the preoperative eGFR at all timepoints. In particular, the postoperative eGFRs at 1, 3, 6, 12, and 24 months were 62.4 ± 19.6 (*p* = 0.021), 60.5 ± 18.5 (*p* = 0.002), 59.5 ± 17.9 (*p* <0.001), 58.8 ± 17.7 (*p* <0.001), and 64.1 ± 20.1 (*p* = 0.001) mL/min/1.73 m^2^, respectively. In the PCA group, the preoperative mean eGFR was 65.7 ± 39.1 mL/min/1.73 m^2^ and the postoperative eGFR was significantly lower than the preoperative eGFR at all timepoints. In particular, the postoperative eGFRs at 1, 3, 6, 12, and 24 months were 55.9 ± 22.2 (*p* <0.001), 54.9 ± 22.5 (*p* <0.001), 51.0 ± 17.9 (*p* <0.001), 51.6 ± 20.7 (*p* <0.001), and 46.4 ± 17.5 (*p* = 0.001) mL/min/1.73 m^2^, respectively.

The changes in % eGFR are shown in Figure 1. Although many patients in the two groups had preserved renal function in the early post-treatment period, the percentage of patients with a % eGFR of <90% increased over time. The univariableanalysis performed to determine factors associated with a >20% reduction in % eGFR at postoperative year 1 showed no significant associations with age, sex, body mass index, hypertension, diabetes mellitus, treatment type (RAPN vs. PCA), and preoperative stage 3a or higher chronic kidney disease (CKD) (Table 4). The univariable and multivariable analyses indicated that an R.E.N.A.L score ≥7 points was a risk factor for >20% reduction in % eGFR at postoperative year 1.

Figure 2 compares the RFS and OS of the patients in the RAPN and PCA groups, excluding two cases, each with benign histology. In brief, the log-rank test revealed that the OS and RFS were slightly worse in the PCA group than in the RAPN group, albeit not significantly (OS, ***p*** = 0.160; RFS, ***p*** = 0.105). Fournier’s gangrene was the cause of death in the RAPN group, whereas prostate cancer, interstitial pneumonia, myocardial infarction, and multiple organ failure were the causes of death in the PCA group. There were no deaths due to RCC in either group.

## 4. Discussion

Cancer-specific survival rates are comparable between PN and RN for localized RCCs, and PN achieves better preservation of renal function than RN, thereby reducing the incidence of metabolic and cardiovascular disorders. Therefore, the best treatment option for localized T1 RCC is PN rather than RN [1]. RAPN is associated with lower blood loss, fewer transfusions and postoperative complications, and lower eGFR decline than PN [12]. In addition, RAPN has been shown to achieve a significantly shorter warm ischemic time and a higher trifecta attainment rate than laparoscopic PN (LPN), indicating its possible advantage in the preservation of renal function [13,14]. Although a high hospitalization cost is a disadvantage, RAPN is superior than other treatment methods, even in frail patients, owing to fewer complications associated with it and shorter hospital stays [15].

Although PN is an appropriate treatment option for localized RCCs as it preserves renal function and has good oncological outcomes, it requires general anesthesia and may not be a feasible option depending on the patient’s general condition, such as impaired cardiac or respiratory function. PCA is a minimally invasive procedure that does not require general anesthesia and can be performed in patients who cannot tolerate general anesthesia. In the present study, 10 of the 49 patients (20.4%) who had undergone PCA could not tolerate general anesthesia. In addition, the patients in the PCA group were significantly older and had a higher rate of cerebrovascular diseases than those in the RAPN group; however, the rates of hypertension, diabetes, and cardiovascular diseases were not significantly different between the two groups. PCA might have been preferred over RAPN due to decreased daily living activities and poor general condition as a result of the advanced age and cerebrovascular disease history.

Studies have reported that the rate of complications is slightly higher for RAPN and LPN than for PCA (Clavien–Dindo grade ≥3 complication rate: 2.6% in RAPN vs. 2.1% in PCA; 3.3% in LPN vs. 1.1% PCA), although the differences were not statistically significant [6,10]. A higher incidence of acute kidney injury with PN than with AT, including radiofrequency ablation and PCA, has also been reported [7]. In the present study, the higher number of patients with a solitary kidney in the PCA group was considered a rational reason for the chosen approach, given the risk of acute kidney injury. There was only one patient with a Clavien–Dindo grade ≥3 complication in the RAPN group, suggesting that RAPN can be safely performed in patients aged ≥70 years. However, PCA may be a safer approach than RAPN because none of the patients who had undergone PCA developed Clavien–Dindo grade ≥3 complications in the present study.

However, a previous study reporting on the 5-year follow-up data of minimally invasive PN (LPN or RAPN) and AT revealed a significantly higher local recurrence rate in the AT group than in the PN group (21.2% vs. 4.7%) [7]. A meta-analysis also reported significantly worse local RFS after AT [16]. In the present study, the RFS was slightly worse in the PCA group than in the RAPN group, albeit not significantly. Local recurrence requires other treatment approaches, such as surgery and AT, and strict follow-up with imaging studies is considered necessary, especially after PCA. RAPN may be a good option for the elderly, considering the low risk of recurrence, consequent re-treatment.

A previous study demonstrated that RAPN is associated with a higher trifecta achievement rate and less deterioration in postoperative renal function compared with LPN [13]. Another study reported that the reduction in eGFR after PCA was not significant [17]. Glomerular filtration rate has been reported to decline with age in elderly patients, even in healthy individuals without any underlying disease [18]. Furthermore, elderly patients have several comorbidities, and the decline in glomerular filtration rate in the elderly is largely due to the presence of comorbidities in addition to normal aging [19]. In the present study, our assessment regarding the preservation of renal function revealed a significant decrease in eGFR in the postoperative period compared with the preoperative period in both the RAPN and PCA groups. The participants in the present study were ≥70 years, and there is a possibility that the observed decline in renal function was due to the high prevalence of comorbidities, such as hypertension and diabetes mellitus. In a previous study, multivariable analysis revealed that a higher R.E.N.A.L score was a significant factor of ≥10% decrease in eGFR at 6 months after RAPN and PCA [6]. In the present study, both the univariable and multivariable analyses revealed that an R.E.N.A.L score of ≥7 was significantly associated with a decline of ≥20% in %eGFR at 1 year after treatment. Therefore, a high R.E.N.A.L score might be a predictor of decreased postoperative renal function.

In patients undergoing PN, initial tumor pathological characteristics and subsequent radiologic imaging results are used to determine treatment success, whereas in patients undergoing AT, radiologic evaluation is used to determine treatment success [20]. Although contrast-enhanced CT or contrast-enhanced MRI is often used to evaluate recurrence after treatment, the use of contrast material may be difficult in patients with impaired renal function due to concerns about contrast-induced nephropathy and nephrogenic systemic fibrosis [21]. The sensitivity of plain CT and plain MRI in detecting locally recurrent lesions is low, which is a concern that should be considered, especially after AT in patients with a high local recurrence rate. The feasibility of follow-up imaging with 111In-girentuximab single-photon emission CT has been reported after PCA for RCC; this approach may improve early detection of residual or recurrent disease and may also be applied in patients with inadequate renal function [22].

The main limitations of the present study were its retrospective study design and small sample size. This study used real-world data, with significantly more patients with T1b tumors in the RAPN group and significantly more patients with a solitary kidney in the PCA group. This difference may have impacted oncological outcomes and post-treatment renal function. Moreover, the follow-up period was short, and only a few patients in the PCA group could be evaluated for renal function at 2 years. Patients with stage 3b or higher CKD were also included, and follow-up imaging might not have accurately assessed recurrence due to the need to reduce or not use contrast media. Because of the lack of clear criteria for treatment selection and significant differences in baseline age, the oncological outcomes and changes in renal function could not be accurately assessed. In addition, a no-treatment follow-up is possible with a longer doubling time in patients with RCC who are older and have more comorbidities; however, this study did not investigate patients on a no-treatment follow-up.

## 5. Conclusions

In elderly patients aged ≥70 years diagnosed with RCC, RAPN was a safe and effective treatment approach with a high nonrecurrence rate. Despite a slightly higher recurrence rate than RAPN, PCA was deemed to be a safe and feasible alternative, especially in patients in whom general anesthesia poses a high risk.

## Figures and Tables

**Figure 1 cancers-14-05843-f001:**
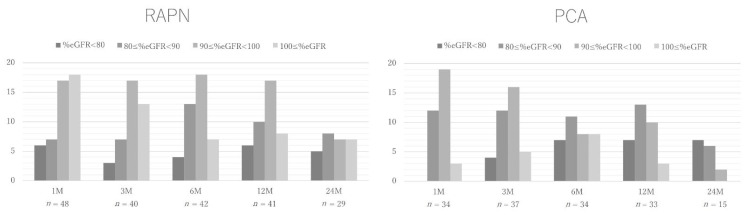
Changes in renal function. Changes in renal function declined in the postoperative period compared with the preoperative period in patients who had undergone robotic-assisted partial nephrectomy (RAPN) and percutaneous cryoablation (PCA). eGFR, estimated glomerular filtration rate.

**Figure 2 cancers-14-05843-f002:**
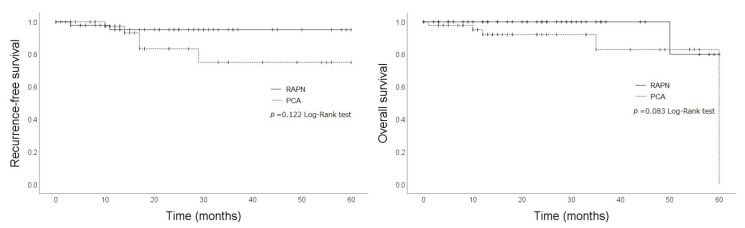
Recurrence-free survival and overall survival. Comparison of recurrence-free survival and overall survival between patients who had undergone robot-assisted partial nephrectomy (RAPN) and those who had undergone percutaneous cryoablation (PCA).

**Table 1 cancers-14-05843-t001:** Patient characteristics and preoperative renal function.

Variable	RAPN	PCA	*p* Value
Patients	50	49	
Median age, years (range)	75 (70–84)	78 (70–91)	0.010
BMI, kg/m^2^	23.1 ± 2.4	23.7 ± 3.8	0.324
Sex, *n* (%)			
Male	34 (68.0)	35 (71.4)	0.711
Female	16 (32.0)	14 (28.6)	
Hypertension, *n* (%)	36 (72.0)	41 (83.7)	0.069
Diabetes, *n* (%)	15 (30.0)	17 (34.7)	0.618
Cardiovascular disease, *n* (%)	7 (14.0)	5 (10.2)	0.563
Cerebrovascular disease, *n* (%)	3 (6.0)	12 (24.5)	0.010
Solitary kidney, *n* (%)	1 (2.0)	8 (16.3)	0.041
Preoperative eGFR, mL/min/1.73 m^2^	65.0 ± 17.6	65.7 ± 39.1	0.334
Preoperative CKD stage			
1	3 (6.0)	9 (18.4)	
2	26 (52.0)	12 (24.5)	
3a	15 (30.0)	15 (30.6)	
3b	6 (12.0)	11 (22.4)	
4	0 (0)	1 (2.0)	
5	0 (0)	1 (2.0)	
CKD stage ≥ 3b	6 (12.0)	13 (26.5)	0.066

BMI, body mass index; CKD, chronic kidney disease; eGFR, estimated glomerular filtration rate; PCA, percutaneous cryoablation; RAPN, robot-assisted partial nephrectomy.

**Table 2 cancers-14-05843-t002:** Reasons for choosing PCA.

Reasons	Details	*n* = 49
Absolute contraindications for RAPN (general anesthesia not available)		
Respiratory diseases	Interstitial pneumonia	3
	Chronic obstructive pulmonary disease	2
	Pyothorax	1
Cardiovascular diseases	Chronic heart failure	2
	Cardiomyopathy	1
	Valvular disease	1
Relative contraindications for RAPN		
Solitary kidney		8
History of previous surgery	Major abdominal surgery	2
	Partial nephrectomy	2
Renal vascular abnormality	Renal artery aneurysm	1
	Renal artery stenosis	1
Difficulty in tumor identification	Polycystic kidney disease	2
	Abnormal renal morphology	1
Patient preference or physician recommendation		
Under treatment for other cancers	Castration-resistant prostate cancer	3
	Hepatocellular carcinoma	2
	Chronic myelogenous leukemia	1
Bilateral or multiple renal tumor	Bilateral	3
	Multiple	1
Extreme age (>85 years old)		4
Patients with comorbidities	Ischemic heart disease	2
	Cerebrovascular disease	2
	Low respiratory function after lung cancer surgery	1
Patients without comorbidities		3

PCA, percutaneous cryoablation; RAPN, robot-assisted partial nephrectomy.

**Table 3 cancers-14-05843-t003:** Tumor features in patients who had undergone RAPN or PCA.

Variable	RAPN	PCA	*p* Value
Maximum tumor diameter, cm	2.7 ± 1.2	2.4 ± 0.8	0.204
Laterality, *n* (%)			
Left	27 (54.0)	29 (59.2)	
Right	23 (46.0)	20 (40.8)	
Clinical T stage			
T1a	42 (84.0)	46 (93.9)	
T1b	8 (16.0)	1 (4.0)	
T3a	0 (0)	2 (4.1)	
R.E.N.A.L. score			
Low (4–6)	29 (59.2)	32 (65.3)	
Moderate (7–9)	20 (40.0)	17 (34.7)	
Highest (10–12)	1 (2.0)	0 (0)	
Pathologic T stage			
T1a	39 (78.0)		
T1b	5 (10.0)		
T3a	4 (8.0)		
Benign	2 (4.0)		
Histologic subtype			
Clear cell	35 (50.0)	38 (77.6)	
Papillary	6 (12.0)	1 (2.0)	
Chromophobe	3 (6.0)	1 (2.0)	
Benign	2 (4.0)	2 (4.1)	
Others	4 (8.0)	0 (0)	
Nondiagnostic	0 (0)	7 (14.3)	

PCA, percutaneous cryoablation; RAPN, robot-assisted partial nephrectomy.

**Table 4 cancers-14-05843-t004:** Results of univariable and multivariable logistic regression analyses for factors associated with the risk of % eGFR <80% at postoperative year 1.

Factor	Patients with %eGFR <80% at Postoperative Year 1			
	Univariable Logistic Regression		Multivariable Logistic Regression	
	*p* Value	OR (95% CI)	*p* Value	OR (95% CI)
Age (≥ 78 years)	0.782	1.190 (0.346–4.097)		
Sex, female	0.077	3.020 (0.886–10.286)	0.079	3.270 (0.873–12.244)
BMI (≥25 kg/m^2^)	0.276	0.463 (0.115–1.854)		
Hypertension	0.478	1.793 (0.357–9.021)		
Diabetes mellitus	0.127	2.579 (0.763–8.715)		
Treatment, RAPN	0.462	0.637 (0.191–2.120)		
Preoperative CKD stage (≥3a)	0.625	1.358 (0.399–4.624)		
R.E.N.A.L. score (≥moderate)	0.008	5.824 (1.581–21.454)	0.008	6.136 (1.596–23.585)

BMI, body mass index; CI, confidence interval; CKD, chronic kidney disease; eGFR, estimated glomerular filtration rate; OR, odds ratio; RAPN, robot-assisted partial nephrectomy.

## Data Availability

The data presented in this study is available within the article.

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
