# Peer review of "Comparison of Clinical Outcomes between Robot-Assisted Partial Nephrectomy and Cryoablation in Elderly Patients with Renal Cancer"

_cancers, 2022, doi:10.3390/cancers14235843_

Round 1

Reviewer 1 Report

This study compares the outcomes of RAPN and PCA for small renal mass and showed that RAPN is a safe and effective treatment for well-selected elderly patients and that PCA is a safe alternative treatment for high-risk patients for treatment and general anesthesia. The study is worthy of publication. Because this is a retrospective analysis of two essentially different treatments, there are differences in the characteristics of the two groups; the RAPN group included more T1b cases and the PCA group included more single kidney cases. This difference may have an impact on the outcome of treatment and the impact of post-treatment renal function and listing it as a limitation is an option.

Author Response

Your comments have been helpful in allowing us to revise our manuscript. Thanks to your comments and suggestions, we believe that our manuscript can be improved. Please re-review our manuscript.

Reply to comment:

Reviewer 1:

Comment: This study compares the outcomes of RAPN and PCA for small renal mass and showed that RAPN is a safe and effective treatment for well-selected elderly patients and that PCA is a safe alternative treatment for high-risk patients for treatment and general anesthesia. The study is worthy of publication. Because this is a retrospective analysis of two essentially different treatments, there are differences in the characteristics of the two groups; the RAPN group included more T1b cases and the PCA group included more single kidney cases. This difference may have an impact on the outcome of treatment and the impact of post-treatment renal function and listing it as a limitation is an option.

Reply to comment:

We have added the following sentence to the limitation of this study, " This study used real-world data, with significantly more patients with T1b tumors in the RAPN group and significantly more patients with a solitary kidney in the PCA group. This difference may have impacted oncological outcomes and post-treatment renal function."

Reviewer 2 Report

This study was reported the comparison of clinical outcomes between RAPN and cryoablation in elderly patients with RCC. The reviewer would like to suggest some critiques to make this paper as follows.

Major revision

1.     First, the author should help of a native English speaker prior to submit the manuscript and make more concise this manuscript. This paper is confusing throughout.

2.     On line 34, what is “with low recurrence”? Recurrence rate? Recurrence-free survival?

3.     On line 34, “Although the recurrence incidence was slightly high,” is inadequate. The authors should be revised this sentence.

4.     The method of citation and bibliography differs from the submission rules.

5.     On line 44, is ultrasonography really a high-resolution imaging?

6.     The endpoint of this study was to evaluate clinical outcomes between RAPN and CPA. The authors should delete several sentences on line 48, 136-141.

7.     On line 53, “While some studies … appropriate treatment selection.” Is unclear.

8.     The authors should describe the follow-up period in this study.  

Author Response

Your comments have been helpful in allowing us to revise our manuscript. Thanks to your comments and suggestions, we believe that our manuscript can be improved. Please re-review our manuscript.

Reply to comment:

Reviewer 2:

Comment: This study was reported the comparison of clinical outcomes between RAPN and cryoablation in elderly patients with RCC. The reviewer would like to suggest some critiques to make this paper as follows.

Major revision

Comment 1.

First, the author should help of a native English speaker prior to submit the manuscript and make more concise this manuscript. This paper is confusing throughout.

Reply to comment 1:

We apologize for any inconvenience caused by the quality of our English. We had already had our manuscript edited by the English editing service prior to the initial submission, but the quality of the English was not sufficient. We believe that the quality of the English has improved since the manuscript was edited again by the English editing service. Please check it out.

Comment 2 and 3.

On line 34, what is “with low recurrence”? Recurrence rate? Recurrence-free survival?

On line 34, “Although the recurrence incidence was slightly high,” is inadequate. The authors should be revised this sentence.

Reply to comment 2 and 3:

We have revised the text of the part you pointed out as follows " The recurrence-free survival and overall survival rates were worse in the PCA group than in the RAPN group, albeit not significantly. RAPN was considered a safe and effective method for treating RCCs in elderly patients. Moreover, although the recurrence rate was slightly higher in the PCA group than in the RAPN group, PCA was deemed to be a safe alternative, especially for treating patients in whom general anesthesia poses a high risk.".

Comment 4.

The method of citation and bibliography differs from the submission rules.

Reply to comment 4:

We have modified our citation and bibliography methods according to the submission rules.

Comment 5.

On line 44, is ultrasonography really a high-resolution imaging?

Reply to comment 5:

We have removed the mention of ultrasonography.

Comment 6.

The endpoint of this study was to evaluate clinical outcomes between RAPN and CPA. The authors should delete several sentences on line 48, 136-141.

Reply to comment 6:

We have removed the text in the section you indicated.

Comment 7.

On line 53, “While some studies … appropriate treatment selection.” Is unclear.

Reply to comment 7:

We have corrected the sentences in the section pointed out as follows: "".

Comment 8.

The authors should describe the follow-up period in this study. 

Reply to comment 8:

We have added the following sentence " The mean follow-up period was 24.3 (2–60) months and 20.1 (2–60) months for the RAPN and PCA groups, respectively."

Round 2

Reviewer 2 Report

None.